# Significance of Diabetic Kidney Disease Biomarkers in Predicting Metabolic-Associated Fatty Liver Disease

**DOI:** 10.3390/biomedicines11071928

**Published:** 2023-07-07

**Authors:** Jaehyun Bae, Byung-Wan Lee

**Affiliations:** 1Division of Endocrinology and Metabolism, Department of Internal Medicine, Catholic Kwandong University College of Medicine, International St. Mary’s Hospital, Incheon 22711, Republic of Korea; baejh0419@gmail.com; 2Division of Endocrinology and Metabolism, Department of Internal Medicine, Yonsei University College of Medicine, Seoul 03722, Republic of Korea

**Keywords:** diabetic kidney disease, metabolic-associated fatty liver disease, biomarker

## Abstract

Metabolic-associated fatty liver disease (MAFLD) and diabetic kidney disease (DKD) share various pathophysiological factors, and epidemiological evidence suggests that these two diseases are associated. Albuminuria and the estimated glomerular filtration rate, which are conventional biomarkers of DKD, are reportedly associated with the risk or severity of MAFLD. Recently, novel DKD biomarkers reflecting renal tubular injury have been introduced to complement conventional DKD markers. In this article, we looked at previous studies that showed an association between MAFLD and DKD, and also reviewed the significance of DKD biomarkers as predictive risk factors for MAFLD.

## 1. Introduction

Non-alcoholic fatty liver disease (NAFLD) is a liver condition that ranges from steatosis to steatohepatitis or fibrosis in the absence of excessive alcohol consumption [1]. The global prevalence of this disease is approximately 30% and increasing [2], and it is currently the most common chronic liver disease worldwide. In patients with type 2 diabetes (T2D), the prevalence of NAFLD is higher than that in the general population and is estimated to be 50–75% [3,4]. Recently, metabolic-associated fatty liver disease (MAFLD) was suggested to more accurately reflect the pathogenesis of chronic liver disease [5].

Chronic kidney disease (CKD) is defined as a sustained reduction in the estimated glomerular filtration rate (eGFR) or evidence of structural or functional abnormalities in the kidney [6]. It is characterized by proteinuria, low eGFR, or both, and its global prevalence is estimated to be >10%, making it one of the leading causes of mortality worldwide [7,8]. Diabetic kidney disease (DKD) is the leading cause of CKD and develops in approximately 40% of patients with T2D [9].

The liver and kidneys are critical for maintaining homeostasis. Because they are affected by systemic changes and have significant effects on other organs, it can be expected that they can influence each other. MAFLD and DKD share many metabolic risk factors and pro-inflammatory pathways [10]. Recently, several studies have reported the association between MAFLD and CKD or DKD [11,12,13,14,15,16,17].

In this article, we review the results of previous studies on the association between MAFLD and DKD, the possible mechanisms linking these diseases, and the association between the biomarkers of DKD and MAFLD.

## 2. Overlap between MAFLD and DKD

### 2.1. Epidemiologic Perspective

As the prevalence of MAFLD and CKD increases, interest in and research on these diseases with increased prevalence has also increased. Accordingly, studies investigating the association between MAFLD and CKD have been actively conducted. Among them, the majority of studies have investigated whether MAFLD affects the occurrence of CKD. In 2014, a meta-analysis of 20 studies (11 cross-sectional and 9 longitudinal) reported that the presence and severity of MAFLD were associated with the risk and severity of CKD [11]. Several years later, a meta-analysis that included nine longitudinal studies reported that MAFLD was associated with an approximately 40% increase in the risk of incident CKD [12]. These analyses defined CKD as an eGFR < 60 mL/min/1.73 m^2^ and/or proteinuria regardless of a cause such as DKD.

To date, no meta-analysis has reported the association between MAFLD and DKD, which is the leading cause of CKD. Several longitudinal studies have been conducted on this topic. In 2008, Italian researchers reported a higher prevalence of DKD in T2D patients with MAFLD in a large cohort study using a cross-sectional design [13]. Shortly thereafter, they published a longitudinal analysis following up 1760 patients with T2D and normal or near-normal kidney function and without overt proteinuria to find the occurrence of DKD, defined as overt proteinuria and/or eGFR < 60 mL/min/1.73 m^2^ for 6.5 years [14]. As a result, MAFLD, diagnosed by liver ultrasound, was associated with an increased incidence of DKD (hazard ratio (HR) 1.69; 95% confidence interval (CI) 1.3 to 2.6; *p* < 0.001). Consistent results have also been reported in another study that recruited an Asian population [15]. In that study, the cumulative incidence of DKD was higher in patients with T2D and MAFLD, and the liver fat content showed a positive relationship with albuminuria and a negative relationship with eGFR. In 2022, Korean researchers demonstrated the relationship between liver fibrosis and DKD [16]. Initially, they failed to show the differential risk of incident DKD between the MAFLD and non-MAFLD groups; however, among the T2D patients with MAFLD, advanced liver fibrosis was significantly associated with DKD (HR 1.75; 95% CI 1.15 to 2.66; *p* = 0.009). Similar results were reported by Chinese researchers [17]. They showed an association between liver fibrosis and DKD incidence and progression in older patients with T2D through both cross-sectional and longitudinal designs. In the case of type 1 diabetes (T1D), another study reported an association between MAFLD and DKD [18]. The result was similar to that of patients with T2D (HR 2.85; 95% CI 1.59 to 5.10; *p* < 0.001). In addition to the studies mentioned above, several longitudinal and cross-sectional studies have reported a significant association between MAFLD and DKD [19,20,21,22,23]. Table 1 summarizes the previous studies that have investigated the association between MAFLD and DKD.

Although considerable evidence has been accumulated, a clear causal relationship between MAFLD and DKD has not yet been identified. Most previous studies adjusted for important risk factors, such as age, sex, body mass index (BMI), glycated hemoglobin (HbA1c), and comorbidities; however, it might be insufficient to adjust for factors shared by MAFLD and DKD, including insulin resistance and abdominal obesity. In addition, studies on the histological findings of MAFLD and DKD are scarce. Studies on changes in DKD according to the progression or improvement of MAFLD, and vice versa, are also lacking. Large-scale studies that address these limitations are required to clarify the association between MAFLD and DKD.

### 2.2. Pathophysiological Mechanisms Linking MAFLD and DKD

Based on the epidemiological evidence described above, MAFLD and DKD are considered to be closely related. Although the precise mechanisms linking MAFLD and DKD are not fully understood, several potential factors and mechanisms may link them.

First, as can be inferred from the names, MAFLD and DKD share broad areas of metabolic dysfunction, such as obesity, insulin resistance, hypertension, dyslipidemia, and diabetes [30]. Insulin resistance is one of the most important pathogenic mechanisms underlying MAFLD and T2D. Insulin resistance, which is also widely known as a fundamental pathological factor in metabolic syndrome, increases lipolysis in adipose tissue, thereby increasing the plasma concentration of free fatty acids [31,32,33]. This induces an excessive accumulation of hepatic triglycerides, resulting in the development and progression of MAFLD. Insulin resistance also results in compensatory hyperinsulinemia, which increases hepatic fatty acid uptake and inhibits β-oxidation, thereby leading to de novo lipogenesis and aggravation of MAFLD. Lipid overload in the liver contributes to the formation of lipotoxic lipids, which leads to the activation of inflammatory responses, mitochondrial dysfunction, and oxidative stress [33]. Progression to hepatic inflammation, that is, the metabolic-associated steatohepatitis state, is not limited to the inflammatory response in the liver but also increases systemic inflammatory responses [34]. Delivery of pro-inflammatory mediators to the kidneys via systemic circulation may act as initiating or aggravating factors for kidney diseases. Inflammatory processes play an important role in the initiation and development of DKD [35,36]. In addition, an increase in oxidative stress leads to a reduction in the level of antioxidant factors in the kidney, such as the Klotho protein [37]. Similar to the other mechanisms discussed below, insulin resistance, MAFLD, and DKD interact with each other, making it difficult to determine the precedence relationship. Insulin resistance is an important pathogenic mechanism of MAFLD that can affect the kidney through various pathways, such as systemic inflammation and oxidative stress. On the other hand, insulin resistance can directly affect kidneys via its pro-inflammatory features and hemodynamic alteration, and vice versa; insulin resistance is known to be a common and very early alteration in CKD, which includes the concept of DKD [38].

Second, expansion and inflammation of the adipose tissue is another common metabolic dysfunction shared by MAFLD and DKD [10]. This encompasses unhealthy diet, obesity, and insulin resistance. Expanded or inflamed adipose tissue releases large amounts of fatty acids into the systemic circulation [10,39,40], which are transported to the liver and cause MAFLD. Inflamed adipose tissue also secretes various pro-inflammatory cytokines such as tumor necrosis factor-α, interleukin-6, resistin, and monocyte chemoattractant protein-1 and increases oxidative stress. MAFLD and systemic inflammation affect the kidneys. Furthermore, renal fat accumulation increases in obese individuals, which may result in local adverse reactions [41,42]. Adiponectin is a representative adipokine that is dysregulated [10,43]. The mechanisms by which adiponectin improves insulin resistance and inhibits reactive oxygen species (ROS) through adenosine monophosphate-activated protein kinase activation have a protective effect against MAFLD and podocyte injury [44]. In contrast, fetuin-A, a hepatokine, induces insulin resistance and suppresses adiponectin production in the adipose tissue [45,46]. The interplay between pro-insulin-resistant fetuin-A and pro-insulin-sensitive adiponectin is considered a common pathogenic mechanism in MAFLD and DKD [10,43,44,47]. Recently, perturbation of the gut microbiota has emerged as a common pathophysiology in MAFLD and DKD [10,30,43,48,49]. Genetic, environmental, and nutritional factors can influence the composition of gut bacteria [50]. In particular, an unhealthy diet adversely alters the intestinal flora. This change, the so-called intestinal dysbiosis, causes the production of gut-derived toxins and promotes their absorption by damaging the intestinal barrier integrity [43]. Increased absorption and accumulation of toxins lead to systemic inflammation, ectopic fat deposition, and insulin resistance. Renal excretion of the toxins and their metabolites can also damage the kidneys [30].

Hyperglycemia can adversely affect both the liver and kidneys in patients with T2D. High concentrations of glucose in the plasma are delivered to the liver and used for de novo lipogenesis [51]. Hyperglycemia induces glomerular hyperfiltration, which initiates and propagates kidney damage in T2D [9,36]. Alteration of glucose metabolism in the kidney also promotes inflammatory responses and fibrotic changes, wherein advanced glycation end products and ROS are the major mediators [36]. Furthermore, in T2D, tubular glucose reabsorption and renal gluconeogenesis are usually increased [52,53]. These changes demand an increased consumption of oxygen, leading to proximal tubule damage, which increases vulnerability to hypoxia [54,55,56,57].

Activation of the renin-angiotensin system (RAS), particularly angiotensin II production, plays an important role in the pathogenesis of MAFLD and CKD [43,58]. In the liver, RAS activation promotes de novo lipogenesis, insulin resistance, and production of pro-inflammatory cytokines [59,60], thereby leading to the development of MAFLD. In the kidney, RAS activation induces renal fat accumulation, inflammatory processes, and vessel constriction, which cause and worsen CKD or DKD [60,61]. In contrast, the progression of DKD also activates RAS; decreased renal blood flow and glomerular filtration rate in the progressive stage of DKD promote renin secretion, thereby activating RAS [62].

Finally, MAFLD and DKD share genetic susceptibilities [30,49]. A representative example is the genetic polymorphism in *PNPLA3* [63], which encodes patatin-like phospholipase domain-containing protein 3 (PNPLA3) and is known to have lipase activity [64]. The *PNPLA3* gene is expressed in the liver and kidneys. *PNPLA3* rs 738,409 polymorphism has been associated with poor renal outcomes and MAFLD severity [63,65,66,67]. Other genetic abnormalities, such as transmembrane 6 superfamily member 2 (*TM6SF2*) and glucokinase regulator (*GCKR*) polymorphisms, have also been reported to be associated with the risk of both MAFLD and CKD [65,68,69,70]. A schematic diagram summarizing the pathophysiological mechanisms linking MAFLD and DKD is shown in Figure 1.

## 3. Biomarkers of DKD and MAFLD

As described in the Section 1, DKD is the leading cause of CKD and one of the most important microvascular complications of T2D [71]. In patients with diabetes, screening for DKD is based on albuminuria and eGFR [72]. Therefore, albuminuria and the eGFR are the representative biomarkers of DKD. We review the association between these conventional DKD biomarkers and MAFLD and then describe the association between tubular markers, introduced as complementary biomarkers for DKD and MAFLD.

### 3.1. Conventional Glomerular Biomarkers: Albuminuria and eGFR

Albuminuria is usually assessed by the random spot urinary albumin-to-creatinine ratio (UACR) [72]. Albuminuria within the normal range is defined as a UACR < 30 mg/g. Diabetic patients with UACR ≥ 30 mg/g in two of three samples collected within 6 months are considered to have DKD. In patients with DKD, most urinary albumin is excreted through the trans glomerular passage [73]. The major mechanism of albuminuria is the impairment of selective permeability of the glomerulus due to glomerular endothelial dysfunction [74]. Therefore, albuminuria primarily reflects glomerular damage.

eGFR is calculated by validated formulas, using serum creatinine, and other variables such as age, sex, and race [75]. The Chronic Kidney Disease Epidemiology Collaboration (CKD-EPI) equation is currently the preferred formulation. In the patient’s laboratory reports, eGFR is usually reported along with the creatinine level, and when it is persistently reduced to <60 mL/min/1.73 m^2^, it is generally interpreted as progressing to a significantly pathological stage [72]. Based on the generally accepted natural course of DKD [76], eGFR decline is believed to occur later than albuminuria.

Considering the guidelines for DKD screening, researchers usually defined DKD as a UACR ≥ 30 mg/g and/or eGFR < 60 mL/min/1.73 m^2^ when they studied the association between DKD and MAFLD, but some studies included total proteinuria or changes in UACR and/or eGFR as variables. In the two meta-analyses introduced in Section 2.1 that showed an association between MAFLD and CKD, CKD was defined as the status covering persistent eGFR < 60 mL/min/1.73 m^2^, albuminuria, and proteinuria on morning urine dipstick. As described in the previous section, no meta-analyses have investigated the association between MAFLD and DKD. However, in 2018, a meta-analysis that summarized the data and estimated the risk of albuminuria among patients with NAFLD showed the results of a subgroup analysis including only patients with diabetes [77]. The subgroup analysis demonstrated no significant association between albuminuria and NAFLD among patients with diabetes [pooled odds ratio (OR) 1.28; 95% CI 0.94 to 1.75], but a significantly increased risk of albuminuria among patients with NAFLD without diabetes (pooled OR 2.25, 95% CI 1.65 to 3.06). However, since then, several studies analyzing a large number of patients with longitudinal or cross-sectional designs have been conducted [16,17,21,25], and most of them have reported a significant association between MAFLD and DKD.

Most of the studies that investigated the association between MAFLD and DKD were designed to evaluate whether MAFLD increased the risk of DKD; the independent variable was MAFLD, assessed by ultrasound, noninvasive indices, or transient elastography (TE), whereas the dependent variable was DKD assessed by albuminuria, proteinuria, or eGFR (Table 1). Studies on whether DKD affected the occurrence or severity of MAFLD and whether DKD biomarkers were related to MAFLD are limited. Several cross-sectional studies have reported that DKD biomarkers can predict the prevalence and severity of MAFLD [21,26,27,28].

In 2020, Chinese researchers conducted a study that recruited 2689 patients with MAFLD and T2D and explored whether a lower eGFR was associated with an increased probability of liver fibrosis [27]. During the inclusion process, MAFLD was determined using ultrasonography. The NAFLD fibrosis score (NFS) was used to evaluate the risk of liver fibrosis. In that study, a negative correlation was found between eGFR and NFS, and the prevalence of liver fibrosis was increased as eGFR quartiles decreased, after adjustment for conventional risk factors (Q4: reference; Q3: OR 1.49; 95% CI 0.82 to 2.71; Q2: OR 1.88; 95% CI 0.97 to 3.67; Q1: OR 2.70; 95% CI 1.36 to 5.37; *p* for trend = 0.002). In the same year, Korean researchers reported an association between albuminuria and MAFLD indices [28]. In that study that enrolled 1108 patients with T2D, the UACR was positively correlated with the NAFLD liver fat score (NLFS), fibrosis-4 index (FIB-4), and NFS. In the logistic regression analysis, albuminuria, defined as a UACR ≥ 30 mg/g, was significantly associated with hepatic steatosis according to NLFS and fibrosis according to NFS, after adjustment for age, sex, and BMI. After further adjustment for HbA1c, homeostasis model assessment of insulin resistance (HOMA-IR), and T2D duration, the association between albuminuria and NLFS remained statistically significant. This study was strengthened by a study published the following year that used TE as an investigative tool for MAFLD; albuminuria was associated with significant steatosis, defined as a controlled attenuation parameter > 302 db/m [26]. In 2022, another research group reported that when patients with T2D were divided into three groups based on a UACR of 30 mg/g and 300 mg/g, the prevalence of MAFLD diagnosed by ultrasound was higher in the group with more severe albuminuria [21].

As described thus far, MAFLD can be correlated with conventional DKD biomarkers, albuminuria, and eGFR. In addition, several cross-sectional studies have reported that conventional DKD biomarkers can predict the risk of MAFLD. As reviewed in Section 2.2, DKD status can affect the liver. Furthermore, albuminuria is closely related to insulin resistance [78,79] and a decreased eGFR generally reflects a later stage of DKD than albuminuria. Considering these points, these results are convincing. Large-scale longitudinal studies are required to clarify the association between conventional DKD biomarkers and MAFLD.

### 3.2. Biomarkers for Renal Tubular Injury

Various factors, such as insulin resistance, hyperglycemia, oxidative stress, and inflammatory processes, play complex roles in the pathogenesis of DKD (Figure 1). Structural changes caused by DKD can also manifest as various alterations in the kidneys, including thickening of the glomerular basement membrane (GBM), mesangial expansion, glomerulosclerosis, and tubular fibrosis or atrophy, etc. [9,80,81,82]. Conventionally, investigations into DKD have focused on glomerular alterations; thickening of the GBM is known to be the earliest structural change in DKD [80], and albuminuria, the most commonly used early DKD biomarker, mainly reflects glomerular damage. However, cases of DKD without significant glomerular alterations have been reported [83,84]; a considerable number of patients with diabetes develop renal impairment without preceding albuminuria.

To overcome this limitation, alterations in renal tubules have gained attention as another important aspect of DKD. Several studies have shown the potential for preceding tubulopathy to cause glomerulopathy in diabetes [85,86], suggesting that detecting tubular damage may potentially be an earlier indicator of DKD compared to identifying glomerular damage through albuminuria assessment in certain groups of patients with T2D. As described in Section 2.2, hypoxic damage to the proximal tubules can be induced in T2D [52,53,54,55,56]. RAS activation causes tubular and glomerular damage [87]. Furthermore, under diabetic conditions, the activities of growth factors, such as transforming growth factor-β, epidermal growth factor, and insulin-like growth factor-1 in renal tubules are increased [82,88], and tubular inflammation and tubulointerstitial fibrosis are induced.

Based on these mechanistic bases and clinical necessity, several biomarkers for tubular injuries, such as N-acetyl-beta-D-glycosaminidase (NAG), neutrophil gelatinase-associated lipocalin (NGAL), liver-type fatty acid-binding protein (L-FABP), and kidney injury molecule-1 (KIM-1) have been introduced as complementary biomarkers for DKD [57]. NAG is a lysosomal enzyme found in the proximal tubule epithelial cells, and urinary NAG is highly sensitive to tubular injury [85,86]. NAG not only reflects DKD status [85,86,89] but is also closely related to various glycemic parameters [90,91]. NGAL is a small protein released from neutrophils and epithelial cells in the renal tubules, lungs, prostate, and digestive tract [92,93]. Owing to its small size, most of the NGAL is filtered through the glomerulus and reabsorbed in the renal tubule [94]. Therefore, in tubular injury, urinary NGAL levels increase. Similar to NAG, urinary NGAL levels are associated with DKD and glycemic control [85,95]. Blood NGAL levels have also been associated with DKD [93,96]. L-FABP is another small protein that is expressed in the proximal tubules and liver [92]. It reflects tubular stress or hypoxia [97] and has been reported to be associated with early stage DKD and its progression [98,99,100,101]. KIM-1 is a transmembrane glycoprotein that is expressed when the proximal tubule is damaged or dedifferentiated after an injury [102]. Its presence in the urine has been reported to be specific for renal tubular damage [102,103], and the association between DKD and urinary KIM-1 has also been reported in several previous studies [95,104]. In addition to these biomarkers, other tubular markers, including retinol-binding protein, alpha-1 microglobulin, and beta-2 microglobulin, have also been reported, and their association with DKD has been presented [57,92]. Because it is difficult to measure each biomarker individually, and no tubular marker has been recognized as a standard biomarker for DKD screening, non-albumin proteinuria (NAP), which can be calculated by subtracting urinary albumin from total protein, has been introduced as an accessible tubular marker that includes a wide range of biomarkers for tubular injury [57]. To summarize the clinical relevance of renal tubular biomarkers, no biomarker of choice represents the standard for detecting renal tubular injuries, such as albuminuria and e-GFR, for detecting glomerular injury and function, respectively.

These tubular markers have received attention relatively recently. Therefore, studies investigating the association between these markers and MAFLD are limited. In the study that showed the association between albuminuria and hepatic steatosis which was introduced in Section 3.1 [28], the researchers also investigated the association between total proteinuria and MAFLD indices. In that study, proteinuria, including urinary albumin and non-albumin proteins, which reflect both glomerular and tubular injury, was defined as spot urinary protein-to-creatinine ratio (UPCR) ≥ 150 mg/g. Proteinuria was associated with liver fibrosis according to the NFS and remained statistically significant even in the final model adjusted for age, sex, BMI, HbA1c, HOMA-IR, duration of T2D, hypertension, alanine aminotransferase level, and total cholesterol level. It can be interpreted that total proteinuria is associated with liver fibrosis, whereas albuminuria is associated with hepatic steatosis. In that study, patients with T2D were divided into three groups based on the UACR and UPCR: non-proteinuria, isolated NAP, and albuminuria groups. Interestingly, patients with isolated NAP showed a significantly increased NFS, comparable to that of the albuminuria group, whereas there was no difference in NLFS. A similar trend was reported in a study using urinary NAG as a biomarker [29]. In patients with T2D, NAG was associated with greater OR for the risk of higher liver fibrosis stage, assessed by TE (F0,1: reference; F2: OR 1.99; 95% CI 1.04 to 3.82; F3,4: OR 2.40; 95% CI 1.52 to 3.80), whereas there was no significant association with the risk of steatosis stage. Taken together, these results suggest that biomarkers of tubular injury may show significant associations with MAFLD, particularly with liver fibrosis.

To date, no studies have used other tubular markers as DKD biomarkers to investigate their association with MAFLD. Because the tubular injury is an important phenotype of DKD and is thought to affect the liver, further studies on the association between these tubular markers and MAFLD are warranted. NGAL and L-FABP have been reported to be related to types of liver injury or liver diseases other than MAFLD [105,106,107,108]. These markers are likely to be associated with MAFLD.

In addition to the tubular markers, other classes of novel DKD biomarkers might be associated with MAFLD. A representative candidate is the serum cystatin-C. Cystatin-C is a low-molecular-weight protein which is filtered freely by the glomerulus [109]. It is known to be less influenced by age, sex, and muscle mass and is more sensitive to renal dysfunction compared to the serum creatinine [109,110]. Serum cystatin-C has also been recognized as an early marker of DKD through considerable studies [111,112]. However, no study has yet reported an association between cystatin-C and MAFLD in patients with diabetes. It is believed that studies on these novel biomarkers will be needed in the future.

## 4. Conclusions

MALFD and DKD share several epidemiologic and pathophysiologic factors and are associated with each other. Accumulating evidence suggests that DKD biomarkers are associated with MAFLD and can predict its occurrence or severity. Because MAFLD is reversible in the early stages, clinicians should check for the presence or severity of MAFLD when treating diabetic patients with DKD. Further studies on the association between novel markers such as tubular markers and MAFLD are required to overcome the limitations of conventional DKD markers.

## Figures and Tables

**Figure 1 biomedicines-11-01928-f001:**
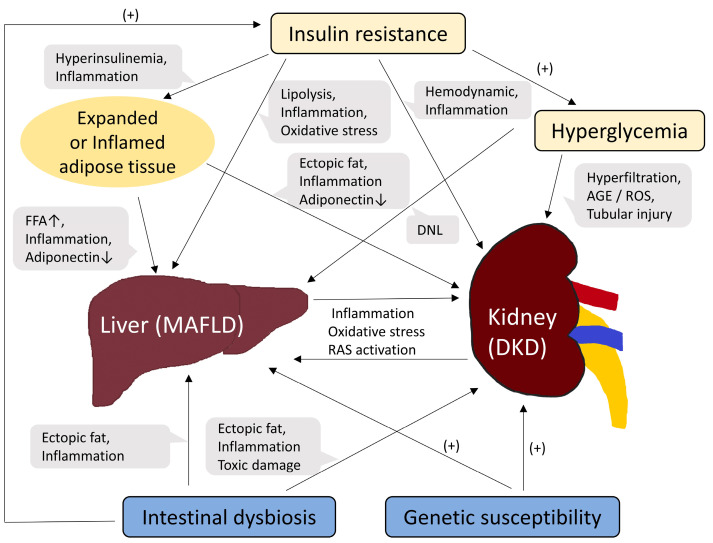
Schematic diagram of common pathogenic mechanisms between MAFLD and DKD. Abbreviations: FFA, free fatty acid; DNL, de novo lipogenesis; AGE, advanced glycation end product; ROS, reactive oxygen species; RAS, renin-angiotensin system.

**Table 1 biomedicines-11-01928-t001:** Previous studies reporting the association between DKD biomarkers and MAFLD.

Design	Patients	Independent Variable	Dependent Variable	Main Finding
Cross-sectional [13]	2103 T2D patients	MAFLD by ultrasound	UACR > 30 mg/g and/or eGFR< 60 mL/min/1.73 m^2^	OR 1.87; 95% CI 1.3 to 4.1
Cohort [14]	1760 T2D patients	MAFLD by ultrasound	UACR > 30 mg/g and/or eGFR< 60 mL/min/1.73 m^2^	HR 1.69; 95% CI 1.3 to 2.6
Cohort [15]	169 T2D matched pairs	MAFLD severity by ultrasound	Incidence of albuminuria (24-h urine albumin > 30 mg)	Increased more in the severe MAFLD group
			Change in eGFR	Decreased more in the severe MAFLD group
Cohort [16]	1729 patients with T2D and MAFLD	FIB-4 index ≥ 2.67	eGFR< 60 mL/min/1.73 m^2^	HR 1.75; 95% CI 1.15 to 2.66
Cohort [17]	1734 T2D patients	FIB-4 index 1.30–3.25	eGFR< 60 mL/min/1.73 m^2^	HR 1.27; 95% CI 1.06 to 1.51
		FIB-4 index > 3.25		HR 2.52 95% CI 1.97 to 3.21
Cross-sectional [17]	3445 T2D patients	FIB-4 index 1.30–3.25	eGFR< 60 mL/min/1.73 m^2^	OR 1.52; 95% CI 1.12 to 2.07
		FIB-4 index > 3.25		OR 3.62; 95% CI 2.26 to 5.80
Cohort [22]	3627 T2D patients	MAFLD by ultrasound	eGFR< 60 mL/min/1.73 m^2^ or ≥2 proteinuria by dipstick	HR 1·30; 95% CI 1·11 to 1·53
Cohort [23]	2057 T2D patients	Liver steatosis (HSI, ZJU)	Albuminuria progression *	HR 1.02; 95% CI 1.00 to 1.03
		Liver fibrosis (BARD)	≥40% eGFR decline	HR 1.12; 95% CI 1.01 to 1.24
Cross-sectional [24]	1763 T2D patients	Liver fibrosis by transient elastography	Incidence of albuminuria (UACR ≥ 3.5 mg/mmol in women and ≥ 2.5 mg/mmol in men)	OR 1.52; 95% CI 1.02 to 2.28
Cross-sectional [25]	2770 T2D patients	Fatty liver index (FLI)	UACR > 30 mg/g	OR 3.49; 95% CI 2.05 to 5.94
			eGFR < 60 mL/min/1.73 m^2^	OR 1.77; 95% CI 1.15 to 2.72
Cross-sectional [26]	100 T2D patients	UACR ≥ 30 mg/g	MAFLD by transient elastography	OR 1.88; 95% CI 1.31 to 2.71
Cross-sectional [21]	1168 T2D patients	UACR ≥ 300 mg/g	MAFLD by ultrasound	OR 2.34; 95% CI 1.20 to 4.56 (vs. UACR < 30 mg/g)
Cross-sectional [27]	2689 T2D patients	eGFR	Hepatic fibrosis by NFS (>0.676)	OR 0.26; 95% CI 0.09 to 0.74
Cross-sectional [28]	1108 T2D patients	UACR ≥ 30 mg/g	Hepatic steatosis by NLFS	OR 1.56; 95% CI 1.01 to 2.40
		UPCR ≥ 150 mg/g	Hepatic fibrosis by NFS	OR 1.55; 95% CI 1.03 to 2.33
Cross-sectional [29]	300 T2D patients	Urinary NAG	Hepatic fibrosis by transient elastography	F2: OR 1.99; 95% CI 1.04 to 3.82
				F3,4: OR 2.4; 95% CI 1.52 to 3.80

* Defined as an increase in the albuminuria stage from normoalbuminuria to microalbuminuria or from microalbuminuria to macroalbuminuria. Abbreviations: T2D, type 2 diabetes; MAFLD, metabolic-associated fatty liver disease; UACR, urine albumin-to-creatinine ratio; eGFR, estimated glomerular filtration rate; OR, odds ratio; CI, confidence interval; HR, hazard ratio; FIB-4, fibrosis-4 index; HSI, hepatic steatosis index; ZJU, Zhejiang University index; NLFS, NAFLD liver fat score; NFS, NAFLD fibrosis score; UPCR, urine protein-to-creatinine ratio; NAG, N-acetyl-beta-D-glucosaminidase.

## Data Availability

No new data were created or analyzed in this study. Data sharing is not applicable to this article.

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
