# Peer review of "Significance of Diabetic Kidney Disease Biomarkers in Predicting Metabolic-Associated Fatty Liver Disease"

_biomedicines, 2023, doi:10.3390/biomedicines11071928_

Round 1

Reviewer 1 Report

This is an interesting manuscript linking two important entities that are rarely considered under one umbrella. Therefore, I find it relevant and sufficiently professionally written for acceptance for publication. 

I wish all the best!

Author Response

Thank you for the favorable comments. 

A slightly revised manuscript will be uploaded that meets other reviewer's suggestion.

Thanks again for your warm comments.

Reviewer 2 Report

The current review manuscript submitted by Jaehyun Bae ad Byung Lee and focused on the biomarkers related to diabetic kidney disease (DKD) and these markers can predict as risk factors for metabolic-associated fatty liver disease (MAFLD). Authors clearly described in the manuscript in regards of several aspects considering the previous supporting evidences based on the clinical data and suggesting the strong correlation between the DKD and MAFLD in terms of epidemiological and pathophysiological factors. Authors concluded that DKD markers in terms of tubular injury (KIM1, NGAL), GFR, albuminuria have strong association with fatty liver disease. This prediction can lead to understand and detect the early stage of metabolic disease and decide to intervention treatment for druggable targets. The manuscript is well written and covered all the aspects previous and ongoing research. The current version can be accepted for publication.

Author Response

(The authors gave the same response as above.)

Reviewer 3 Report

Biomedicines

biomedicines-2480187

Title: Significance of diabetic kidney disease biomarkers in predicting metabolic -associated fatty liver disease

This review summarized that MALFD and DKD share several epidemiologic and pathophysiologic factors and are associated with each other. Accumulating evidence suggests that DKD biomarkers are associated with MAFLD and can predict its occurrence or severity. Because MAFLD is reversible in the early stages, clinicians should check for the presence or severity of MAFLD when treating diabetic patients with DKD. Further studies on the association between tubular markers and MAFLD are required to overcome the limitations of conventional DKD markers.

These points are highly significant and logical. Particularly, Figure 1, which presents a schematic diagram illustrating the common pathogenic mechanisms between MAFLD and DKD, effectively demonstrates several shared pathways in both conditions. It provides readers with a clear understanding of the importance of addressing modifiable risk factors for DKD, which can potentially contribute to the prevention or reversal of MAFLD.

However, we have some questions and recommendations regarding this review:

Comments list

(Q.1) In Figure 1, there are several abbreviations used, such as DNL. It is recommended to provide the full names of these abbreviations to enhance clarity and understanding for readers.

(Q.2) In the section of “3.1. Conventional glomerular biomarkers: albuminuria and eGFR”, we suggest revising the statement "Normal albuminuria is defined as a UACR < 30 mg/g" to "Albuminuria within the normal range is defined as a UACR < 30 mg/g." This revision clarifies that we are referring to albuminuria that falls within the normal range.

(Q.3) In the section of “3.1. Conventional glomerular biomarkers: albuminuria and eGFR”, The statement "Several cross-sectional studies have reported that DKD biomarkers can predict the prevalence and severity of MAFLD" may be revised to provide more clarity (or attach associated references).

(Q.4) In the section of “3.2. Biomarkers for renal tubular injury”, we suggest revising the statement of “detecting tubular damage might be faster than detecting glomerular damage as assessed by albuminuria in identifying DKD in certain groups of patients with T2D.” to “detecting tubular damage may potentially be an earlier indicator of DKD compared to identifying glomerular damage through albuminuria assessment in certain groups of patients with T2D."

(Q.5) Another well-known biomarker for diabetic kidney disease (DKD) is "Cystatin C." We recommend summarizing some relevant studies on this biomarker in MAFLD, too.

The quality of the English language is considered acceptable, as it allows for easily readable communication.

Author Response

We would like to thank the Reviewer 3 for favorable comments and criticism, which significantly helped us to improve the quality of this manuscript. In response to the comments, we have revised manuscript. We would like to reveal that we have made changes in the manuscript according to the reviewers’ comments. All changes in the revised manuscript are coloured in red, but not as tracked changes for your convenience.

<Comment list>

(Q.1) In Figure 1, there are several abbreviations used, such as DNL. It is recommended to provide the full names of these abbreviations to enhance clarity and understanding for readers.

Response) Thank you for the important suggestion. We have added the abbreviations section as the legend of Figure 1.  

(Q.2) In the section of “3.1. Conventional glomerular biomarkers: albuminuria and eGFR”, we suggest revising the statement "Normal albuminuria is defined as a UACR < 30 mg/g" to "Albuminuria within the normal range is defined as a UACR < 30 mg/g." This revision clarifies that we are referring to albuminuria that falls within the normal range.

 Response) Thank you for the valuable suggestion. We have changed the statement according to your recommendation.

(Q.3) In the section of “3.1. Conventional glomerular biomarkers: albuminuria and eGFR”, The statement "Several cross-sectional studies have reported that DKD biomarkers can predict the prevalence and severity of MAFLD" may be revised to provide more clarity (or attach associated references).

Response) Thank you for the significant opinion. Following paragraph which started with the sentence “In 2020, Chinese researchers~” is the examples of the several cross-sectional studies that have reported that DKD biomarkers can predict the prevalence and severity of MAFLD. We added the references of them to the statement as your recommendation. Thank you again for your meticulous review.

(Q.4) In the section of “3.2. Biomarkers for renal tubular injury”, we suggest revising the statement of “detecting tubular damage might be faster than detecting glomerular damage as assessed by albuminuria in identifying DKD in certain groups of patients with T2D.” to “detecting tubular damage may potentially be an earlier indicator of DKD compared to identifying glomerular damage through albuminuria assessment in certain groups of patients with T2D."

Response) Thank you for the important suggestion. We have changed the statement according to your recommendation.

(Q.5) Another well-known biomarker for diabetic kidney disease (DKD) is "Cystatin C." We recommend summarizing some relevant studies on this biomarker in MAFLD, too.

Response) Thank you for the valuable suggestion. Cystatin-C has been reported as an early marker for CKD and DKD. However, there was no study which reported the association between cystatin-C and MAFLD in patients with diabetes. We have added the paragraph which introduced the cystatin-C at the end of the section 3.2, as follows;

In addition to the tubular markers, other class of novel DKD biomarkers might be associated with MAFLD. A representative candidate is the serum cystatin-C. Cystatin-C is a low-molecular-weight protein which is filtered freely by the glomerulus [109]. It is known to be less influenced by age, sex, and muscle mass and is more sensitive to renal dysfunction, compared to the serum creatinine [109,110]. Serum cystatin-C has also been recognized as an early marker of DKD through considerable studies [111,112]. However, no study has yet reported an association between cystatin-C and MAFLD in patients with diabetes. It is believed that studies on these novel biomarkers will be needed in the future.

(You can also check the comments list and responses in the word file which is provided as an attachment.)

Thank you again for your valuable comments.
